# A Quinacridone-Diphenylquinoxaline-Based Copolymer for Organic Field-Effect Transistors

**DOI:** 10.3390/polym11030563

**Published:** 2019-03-26

**Authors:** Yong Jin Jeong, Jeong Hyun Oh, Ho Jun Song, Tae Kyu An

**Affiliations:** 1Department of Materials Science and Engineering, Korea National University of Transportation, Chungju 27469, Korea; il5range@gmail.com; 2Department of Polymer Science & Engineering, Korea National University of Transportation, Chungju 27469, Korea; i0055735@naver.com; 3Research Institute of Sustainable Manufacturing System Intelligent Sustainable Materials R&D Group, Korea Institute of Industrial Technology, Chungcheongnam-do 31056, Korea; 4Department of IT Convergence, Korea National University of Transportation, Chungju 27469, Korea

**Keywords:** organic field-effect transistor (OFET), organic semiconductor, quinacridone, thermal annealing, channel resistance, polymeric semiconductor

## Abstract

In this work, we characterized poly(quinacridone-diphenylquinoxaline) (PQCTQx). PQCTQx was synthesized by a Suzuki coupling reaction and the synthesized PQCTQx was used as a polymeric semiconducting material in organic field-effect transistors (OFETs) to research the potential of using quinacridone derivatives. The measured field-effect mobility of the pristine PQCTQx film was 6.1 × 10^−3^ cm^2^/(V·s). A PQCTQx film heat-treated at 150 °C exhibited good field-effect performances with a hole mobility of 1.2 × 10^−2^ cm^2^/(V·s). The improved OFET behaviors resulting from the mild thermal treatment was attributed to improved packing of the molecules in the film, as determined using X-ray diffraction, and to decreased channel resistance.

## 1. Introduction 

Research into organic field-effect transistors (OFETs) has been committed to the advancement of organic semiconducting materials and device engineering technologies in the last decades [1,2,3,4,5]. Current investigations of OFETs are aiming to rapidly meet the rising demand for large-area or flexible sensors and displays, and driving elements for RF identification tags, as well as many other lower-cost functional devices [6,7]. While OFETs generally work best in applications that do not typically need high-performance semiconducting materials, such as single crystalline and polycrystalline silicon, high field-effect mobility and good device-to-device reproducibility (which rely partially upon the structural consistency of the OFET film) are nonetheless significant even for inexpensive electronic circuits now that the numerous transistors in these devices have to function all together [8]. To enhance OFET performance, one of the most important considerations is developing molecular design strategies aimed at enhancing the packing of the molecules in the OFET. For this purpose, various organic semiconductors have been synthesized and analyzed. The greatest advantage of using organic molecules as semiconductors is that it is easy to make planar conjugated structures out of these molecules, and such structures strongly form π-π interactions and bring about fast charge transport [9].

Polymeric semiconductors, in particular, show superior mechanical flexibility and are amenable to solution processes for large areas in forming a uniform film morphology, and are hence regarded as promising candidates for active semiconducting materials in OFETs [10]. Various kinds of polymeric semiconductors have been developed, but quinacridone (QC)-based polymers have drawn particularly significant attention for use in organic electronic devices, because these polymers have a well-ordering structure and are able to self-assemble [11]. The quinacridone moiety has been widely studied as electron-withdrawing units and red-violet pigments in donor-acceptor (D-A) copolymers for organic solar cells and light-emitting diodes [12]. The application of a quinacridone-based polymer in light-emitting diodes was first reported by the Wang group [13].

Copolymers, including quinacridone units, may be promising candidates for using semiconducting materials in OFETs now that the quinacridone unit has a planar and simple molecular structure that can easily participate in π-π interactions, and hence lead to efficient charge transport. The Fu group reported the OFET performance of quinacridone–benzothiadiazole-based D-A copolymers and showed their hole mobility values in each case to be 0.3 cm^2^/(V·s) [12]. Recently, our group also reported the use of poly[quinacridone-alt-quaterthiophene] (PQCQT) and poly(quinacridone-quinoxaline) (PQCQx) as channel materials in OFETs with hole mobilities of 2.0 × 10^−2^ and 1.4 × 10^−3^ cm^2^/(V·s) when using PQCQT and PQCQx [14,15]. As the field of developing quinacridone–based D-A copolymer semiconductors have progressed, it is necessary to investigate thermal stability of the quinacridone-based D-A polymer crystals and the effect of thermal annealing process on the lateral charge transports that can be evaluated from the field-effect mobility and channel resistance in the OFETs.

In this work, we characterized poly(quinacridone-diphenylquinoxaline) (PQCTQx), whose structure is shown in Figure 1, for applications in OFETs. Although PQCTQx has been investigated by the Moon group for employment in organic solar cells [16], it has not, to the best of our knowledge, been applied to a semiconducting material in OFETs. The PQCTQx OFET device exhibited p-type performance of hole mobility values of 6.5 × 10^−3^ cm^2^/(V·s) and 1.3 × 10^−2^ cm^2^/(V·s), resulting from pristine and thermally heat-treated PQCTQx films, respectively. The different OFET behaviors resulting from the pristine and heat-treated PQCTQx films we prepared were explained by the results of X-ray diffraction (XRD) and channel resistance experiments.

## 2. Experimental

### 2.1. Materials, Device Fabrication

The PQCTQx was synthesized by a Suzuki coupling reaction, as reported formerly [16] (shown as Scheme 1). The synthesized PQCTQx polymer was found to have a number-average molecular weight (M_n_) of 85.0 kg/mol, and a polydispersity index (PDI) of 7.17, by employing gel permeation chromatography method with tetrahydrofuran at 40 °C. The solution-processed PQCTQx film’s electrical properties were investigated with the film in a top-contact/bottom-gate OFET configuration by using a 300-nm-thick SiO_2_ dielectric on an extremely doped n-Si substrate, which functioned as the gate electrode. The SiO_2_ dielectric was treated with an octadecyltrichlorosilane (ODTS) monolayer in the toluene solution for 90 min at room temperature. Solutions of the PQCTQx were made at a concentration of 0.2 wt % in chloroform and heated at 50 °C for 30 min, and then filtered by using a 0.2-µm-pore-sized polytetrafluoroethylene membrane syringe filter. The polymer film was formed by conducting the spin coating method for 60 s at 2000 rpm. On top of the semiconductor layers (100 nm), gold source were deposited and electrodes were drained by using a patterned tungsten shadow mask over the PQCTQx layer. When measured, the channel length (L) and width (W) were 50 µm and 1000 µm, respectively. The OFET devices were heat-treated at 150 and 200 °C for 10 min. At this time, the heat treatment proceeded under a nitrogen atmosphere.

### 2.2. Electrical Characterization of the OFET Devices

Under a nitrogen atmosphere, PQCTQx OFET devices were analyzed by a Keithley 4200 SCS (Keithley Instruments, Cleveland, USA) at room temperature. Field-effect mobilities were calculated in the saturation regime from the slope of a line fitting a plot of the square root of the source–drain current (*I_DS_*) versus the gate voltage (*V_G_*); the line fitting was obtained from the equation *I_DS_* = (*WC_i_*/2L)*μ*(*V_G_* − *V_th_*)^2^, where *C_i_* is the capacitance per unit area of the dielectric (10 nF/cm^2^), *μ* is the filed-effect mobility, and *V_th_* is the threshold voltage. The total width-normalized channel resistances (*R*_total_*W*) were obtained from the inverse slope of the *I_DS_*-*V_D_* curves in the linear regime using output characteristics. We measured *R*_total_*W* using the output curves with *V_G_* of −80 V applied.

### 2.3. Morphological Characterization

XRD experiments were conducted by X-rays with an energy level of 11.57 keV at the 5A beamline of the Pohang Accelerator Laboratory (PAL), Pohang, Korea. For XRD studies, the thin-film samples were prepared by spin coating method at 2000 rpm with the 0.2% chloroform solution on an ODTS-treated Si wafer to copy the OFET device making process. To confirm the effect of thermal annealing, the some of the deposited films were heat treated at 150 and 200 °C.

## 3. Results and Discussion

The transfer characteristics of the prepared PQCTQx-based OFETs are shown in Figure 2. The devices exhibited the formal p-channel transfer characteristics and the heat treatments enhanced the field-effect mobility (Table 1). The saturation field-effect mobility of the pristine PQCTQx film was extracted from the slope of the matched plot in Figure 2 to be 6.1 × 10^−3^ cm^2^/(V·s) with an on/off ratio of 1.7 × 10^5^. For the heat-treated film at 150 °C, the field-effect mobility was greater, with a value of 1.2 × 10^−2^ cm^2^/(V·s), and with an on/off ratio of 3.0 × 10^5^. The mobility value decreased, however, when the temperature of heat treatment was increasingly raised by 50 °C; in the heat-treated film at 200 °C, the field-effect mobility was indicated to be 6.0 × 10^−3^ cm^2^/(V·s) with an on/off ratio of 5.1 × 10^4^. These different OFET behaviors may have been due to structural differences between the as-cast and heat-treated films shown by the XRD analysis and channel resistance measurements described below.

We investigated the molecular-level structures of the PQCTQx films by using XRD. As shown in Figure 3a, the pristine PQCTQx film showed a weak (010) peak, in the out-of-plane direction of the film, and this peak was attributed to molecular stacking of π-π interaction. After heat treating at 150 °C, the intensity of the (010) peak increased, and indicated a π-π distance of 4.0 Å. The XRD pattern in Figure 3b revealed that the diffraction peaks seemed to appear along the in-plane direction of the pristine film, though they were not clear. After annealing at 150 °C, (100) and (010) peaks were observed corresponding to a d-spacing of 23.6 Å, showing a face-on interchain ordering. Although an edge-on structure would be expected to promote faster carrier transport along the π-π stacking path in the OFET structure than would a face-on structure, a face-on structure is more suitable than an amorphous structure [17]. Previous work in our group has investigated PQCQT as the active material in OFET devices [14]. A comparison of the crystalline structures of PQCQT and PQCTQx in film state revealed that the crystalline structure of PQCTQx was not as good as that of PQCQT for charge transport in OFET devices [17,18]. Because the diphenylquinoxaline unit in PQCTQx is in bulky groups and generates steric hindrance, edge-on arrangement may become difficult [19,20]. Increasing the annealing temperature by an additional 50 °C yielded decreased intensities of the observed peaks along both (100) and (010) directions. The relatively high intensities of the PQCTQx XRD peaks subjected to thermal annealing at 150 °C suggested that this PQCTQx film was highly crystalline. The considerably crystalline structure of PQCTQx apparently induced, as described above, favorable intermolecular self-assembled interactions and efficient charge carrier transport in the PQCTQx OFETs.

Channel resistance data were also collected for the various OFET devices. The channel resistances (*R*_total_*W*) were obtained from the inverse slope of the output curves in the linear regime at *V_G_* = −80 V. As shown in Figure 4, the channel resistance value resulting from the as-cast PQCTQx film was measured to be 21.5 MΩcm, and the values resulting from the films annealed at 150 °C and 200 °C were measured to be 16.7 MΩcm and 25.6 MΩcm, respectively. The relatively low channel resistance value resulting from the film heat treated at 150 °C led to the better field-effect mobility in its device [21,22].

## 4. Conclusions

In this research, a semiconducting PQCTQx polymer based upon a diphenylquinoxaline and quinacridone unit was applied to OFET devices. A mild thermal annealing treatment improved the PQCTQx OFET performance. The PQCTQx film annealed at 150 °C exhibited a more face-on ordering structure, enhanced field-effect mobility, with a value of 1.2 × 10^−2^ cm^2^/(Vs), and a lower channel resistance, with a value of 16.7 MΩcm, than did the as-cast film or the film heat treated at 200 °C.

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
