# Peer review of "A Quinacridone-Diphenylquinoxaline-Based Copolymer for Organic Field-Effect Transistors"

_polymers, 2019, doi:10.3390/polym11030563_

Round 1

Reviewer 1 Report

In this work, synthesized poly(quinacridone-diphenylquinoxaline) (PQCTQx) was used as a polymeric semiconducting material in organic field-effect transistors (OFETs) to research the potential of using quinacridone derivatives.

Please find below my comments and suggestions:

     1. Introduction

·       The topic is presented clearly, and references given here are appropriate. The authors have already done some research in this field and references of previous works have been included in this document and cited properly (References 14-16).

·       I suggest doing Figure 1 a bit larger to be able to see clearly molecular structure. 

2. Experimental

Sample preparation is described with enough detail in this section, including also references from previous works.

Referring to section 2.1., the polydispersity index is a bit high (PDI=7.17). Could you comment on the possible effects on the final response of the material?

In Figure 2 (a) the log scale on the right-hand side is a bit small to see the divisions in the y-axis.

In section 2.3., sample preparation for XRD and AFM experiments is described. However, AFM experiments are not included in the present document. Is there any missing information? The following sentence was written: “For XRD and AFM studies, the thin film samples were prepared by spin coating method at 2000 rpm with the 0.2% chloroform solution on an ODTS-treated Si wafer to copy the OFET device making process.” It would be worth having the results of AFM experiments.

Based on these comments and suggestions, I recommend the paper for publication after minor changes.

Author Response

I attached the file of the response to the reviewer's comments.

Reviewer 2 Report

This manuscript contains the characterization ofpoly(quinacridone-diphenylquinoxaline)(PQCTQx) as a polymeric semiconducting material in organic field-effect transistors (OFETs)The background as well as the results are well presented, and the authors take great care only to limitthe impact of D−A interaction on field-effect mobilityproperties of (PQCTQx) as well as evaluation of improvement ofOFET behaviors bythe mild thermal treatment.Thus, this manuscript could be interesting for the readers in this field of material sciencesas well as electronic devices. Also this paper is easy to read and could well serve as the basis for a devel­opment of these fields.

Comments

1.    The authors explained that the improvement of OFET behaviors of PQCTQx by the mild thermal treatment was attributed to the improved packing of the molecules in the film as determined only using X-ray diffraction and channel resistance experiments. The additional evidences for these findings are required in the text, such as the results of microscopic analysis except XRD and AFM studies.

2.    Although synthesis and evaluations of PQCTQx in organic solar cells has been investigated by the Moon group (reference 16), it is helpful for readers to introduce briefly their synthetic strategy and final goal for their research projects. 

3.    Additional description of the Figures and Table supplement are required.

4.    In Figure 1, a symbol of degree of polymerization in PQCTQx polymer is required, such as “n”. And remove the “(a)”.

I recommend this manuscript for publication inPolymers after minor revisions described above.

Author Response

I attached the file of the response to the reviewer's comments.

Polymers EISSN 2073-4360 Published by MDPI AG, Basel, Switzerland RSS E-Mail Table of Contents Alert
Back to Top